# Changes in Availability and Affordability on the University Food Environment: The Potential Influence of the COVID-19 Pandemic

**DOI:** 10.3390/ijerph21121544

**Published:** 2024-11-21

**Authors:** Patrícia Maria Périco Perez, Maria Eduarda Ribeiro José, Isabella Fideles da Silva, Ana Cláudia Mazzonetto, Daniela Silva Canella

**Affiliations:** Institute of Nutrition, Rio de Janeiro State University (UERJ), R. São Francisco Xavier, 524-Maracanã, Rio de Janeiro 20550-013, RJ, Brazil; patriciapericoperez@gmail.com (P.M.P.P.); mariaeduardarj11@hotmail.com (M.E.R.J.); isafideles1@gmail.com (I.F.d.S.); ac.mazzonetto@gmail.com (A.C.M.)

**Keywords:** organizational food environment, campus, middle-income countries, food and nutrition security

## Abstract

Background: The COVID-19 pandemic has had an impact on the eating habits of the general population, among other reasons, because it has affected access to commercial establishments since some of them closed. This study aimed to describe potential changes that occurred between 2019 and 2022 in the availability and affordability of food and beverages in the food environment of a Brazilian public university. Methods: Cross-sectional and descriptive study conducted at a public university located in Rio de Janeiro, Brazil. Audits were carried out in all establishments selling food and beverages at the university’s main campus in 2019 and 2022. Descriptive analysis with frequencies and means was carried out and the 95% confidence intervals were compared. Results: Over the period, there was a decrease in the on-campus number of establishments, dropping from 20 to 14, and ultra-processed foods became more prevalent. In general, the decrease in the number of establishments led to a reduction in the supply of fresh or minimally processed foods and beverages, and higher average prices were noted. Conclusions: The pandemic seems to have deteriorated the availability and the prices of healthy food in the university food environment, making healthy choices harder for students and the university community.

## 1. Introduction

The food environment encompasses the physical, economic, political, and socio-cultural context in which consumers interact with the food system, and it influences their decisions on the purchase, preparation, and consumption of food [1,2]. Some theoretical approaches highlight the organizational food environment [3,4,5], defined as the place intended for the sale or supply of food to workers, students, and members of institutions and organizations. This concept encompasses spaces such as schools, universities, companies, public services, hospitals, prisons, and civil society associations, including their respective food centers, such as canteens, kiosks, and food vending machines [4]. The university food environment combines education and work into one single space and gathers together students and workers of varying ages and income levels. When students—usually young people—enter university, they often take responsibility for managing their own housing, food, budget, and time.

Several studies have shown that the university food environment discourages healthy eating and/or stimulates unhealthy eating practices, owing to the low nutritional quality of the foods available in food establishments [6,7,8,9,10,11,12,13,14]. In this sense, the implementation of strategies, e.g., increasing the availability and diversity of healthy foods at affordable prices in food establishments [6,9,10,13,14,15,16,17], encouraging the consumption of these foods through special offers [6,7,9,10], and reducing the advertising of unhealthy foods [6,9,10], among other measures, seem to be effective in promoting healthier food choices.

The COVID-19 pandemic has had a significant impact on food intake by the general population; one of the reasons is that it affects access to commercial establishments, including those located at universities [18]. Restrictions such as physical distancing and temporary lockdowns have caused a sharp decline in customer flow, resulting in significant financial impact for the owners of these establishments, and, in some cases, they were permanently closed. This scenario also caused a decrease in purchasing power, leading many individuals to opt for ultra-processed foods more often. This choice was influenced by the increased promotion of these products by several companies, based on extended durability and ease of storage [19,20,21]. By recognizing the central importance of physical and financial access to food in food practices and the negative impact of the COVID-19 pandemic on this situation, the aim of this study was to describe potential changes that occurred between 2019 and 2022 in the availability and affordability of food and beverages in the food environment of a Brazilian public university.

## 2. Materials and Methods

### 2.1. Study Design

This is a cross-sectional and descriptive study conducted at a public university located in the city of Rio de Janeiro. The university has 11 campuses, but this study focuses on the main campus, whose major building has an urban and vertical design, and 12 floors. This campus stands out for a high flow of people: approximately 32 thousand/day, around 75% of the university community. In addition, the campus offers a wide variety of degree programs, covering the areas of Technology and Sciences, Biomedicine, Social Sciences, and Education and Humanities. Academic activities on this campus extend from 7 a.m. to 11 p.m. hours, Monday to Friday, and some of them also take place on Saturdays.

### 2.2. Data Collection and Audit Tool

Audits were carried out by trained researchers in all establishments (commercial and non-commercial) selling food and beverages on campus in October 2019 (Collection 1) and November 2022 (Collection 2). In both periods, the activities of the establishments can be considered stable since the classes were regularly happening at that time. The months represented an equivalent moment in the semesters, the middle, and the same season, which contributed to the comparison. It is relevant to mention that in 2022 the face-to-face classes returned in March, eight months before the data collection. This time difference between the university’s reopening and the data collection was important because the university took some time to reorganize its activities and structures after a long period of most of the activities happening online.

The checklist developed and used for auditing (http://www.observatoriodeobesidade.uerj.br/?p=3294, accessed on 19 November 2024) was based on the theoretical approach proposed by Glanz et al. [3] and Caspi et al. [15]. It took industrial food processing [22] into account, and content reliability and validity were evaluated by Franco et al. [10]. This checklist consists of 38 questions divided into 7 blocks: Block 1—characterization of the establishment; Block 2—observation of the environment; Block 3—information; Block 4—beverages, foods, and preparations; Block 5—convenience items; Block 6—prices and promotions; and Block 7—advertisements.

The present study analyzed data on the dimensions of availability and affordability (related to financial accessibility) of food and beverages [15]. Thus, to characterize the university food environment and evaluate the influence of the pandemic on this environment, the following variables were considered:–Description of establishments: number of establishments; proportion of establishments according to the type of food served (snack, meal, convenience items); payment method.–Characterization of the supply of food, beverages, and preparations: fresh or minimally processed foods (5 items); fresh or minimally processed beverages (2 items); ultra-processed foods (10 items); and ultra-processed beverages (10 items).–Promotion of food, beverages, and preparations: the possibility of replacing items (4 options); change of portion size (increase or reduction in size) for an equal, higher, or lower price; the presence of “combos/sales offer”.–Price of food, drinks, and preparations: a record of the lowest and highest price of meals (2 items), food (12 items), and beverages (10 items).

### 2.3. Data Analysis

The databases were exported into an Excel spreadsheet and analyzed in the Stata SE software, version 14.2 (Stata Corp., College Station, TX, USA).

Descriptive analysis of data was performed using absolute and relative frequency measurements in the case of categorical variables, presenting confidence intervals (95% CI), and measures of central tendency (means) and dispersion (minimum and maximum values), in the case of continuous variables.

The comparison between confidence intervals (95% CI) showed significant differences. The absence of overlap between the intervals was assumed as a significant difference, considering the significance level of 5%.

### 2.4. Ethical Aspects

The present study was approved by the Research Ethics Committee of the Pedro Ernesto University Hospital from the Rio de Janeiro State University (Process 5,774,536). All existing establishments agreed to participate and authorized observation and data collection.

## 3. Results

In 2019, the University had 20 establishments selling food, beverages, and food preparations/meals, while in 2022 this number decreased to 14. In both periods, one of the establishments was the university restaurant, a non-commercial establishment. Despite this reduction in the number of establishments, it is remarkable that most of the items served in these places continued to be convenience items (90.0% in 2019 and 85.7% in 2022) and snacks (75.0% in 2019 and 78.6% in 2022), even though there was no statistically significant difference between the years (Table 1).

As for payment methods, the scenario remained practically unchanged, with cash (95.0% in 2019 and 92.9% in 2022) and debit cards (90.0% in 2019 and 92.9% in 2022) being the most common payment methods. Interestingly, there was a change in the acceptance of credit cards as a payment method between 2019 and 2022. In 2019, only half of the establishments (50%) adopted this payment method, but in 2022, it was accepted in all establishments (100%). Other payment methods, such as Pix, an instant payment system developed by the Central Bank of Brazil in 2020 (Pix is a term selected by the Central Bank of Brazil and not an acronym), gained prominence in 2022 (Table 1).

There was a reduction in the absolute number of establishments that offered fresh or minimally processed foods in 2022, i.e., less than 1/3 of the establishments, although there was no statistical difference. However, the most prevalent food items continued to be legumes (40.0% in 2019 and 28.6% in 2022), followed by raw vegetables (35.0% in 2019 and 28.6% in 2022). The offer of fresh or minimally processed drinks was reduced between the years 2019 and 2022. There was a change in the sale of water. In 2019, most establishments (90%) sold water; however, by 2022, there was a decrease with only 78.6% selling water (Table 2).

On the other hand, there was a predominance of establishments selling ultra-processed foods in the study period. Although the offer of ultra-processed foods was reduced in 2022, they are still found in more than 50% of establishments, except for whole cookies (21.4%). In 2019, the most frequent items were candies (90.0%), sweet biscuits with filling, biscuits and/or “packet” salted snacks (85.0%), and other sweets (85.0%). In 2022, the most available ultra-processed foods were fried/baked snack foods (78.6%), candies (78.6%), bonbons and chocolate bars (71.4%), and other types of sweets (71.4%) (Table 2).

The most available ultra-processed beverage options remained the same throughout the period: ready-to-drink tea (90.0% in 2019 and 78.6% in 2022), guarana drink (90.0% in 2019 and 64.3% in 2022), and soft drinks (85.0% in 2019 and 78.6% in 2023) (Table 2). There were no alcoholic beverages in any of the establishments in 2019, but they were sold in one establishment in 2022.

In 2022, there were no establishments that replaced food items or changed portion sizes. Instead, 70% of the establishments in 2019 and 64.3% in 2022 predominantly adopted the practice of providing combo offers or sales offers (Table 3). In 2019, 15% of the establishments offered exclusive promotional prices to university students, but this proportion decreased to 7% in 2022.

The average price of most evaluated foods increased between 2019 and 2022, except for whole cookies (R$3.30–R$2.10) and water (R$2.33–R$2.30). A notable highlight is the average price of fruit or fruit salad, which almost tripled, from R$5.66 in 2019 to R$15.00 in 2022. The price of meals in the university restaurant remained the same (R$2.00 for students) (Table 3).

## 4. Discussion

The evaluation of the food environment of the main campus of a Brazilian university located in Rio de Janeiro, comparing the period before (2019) and after (2022) the COVID-19 pandemic, showed a decrease in the number of establishments that sold food at the university. In both periods, those who sold convenience items and snacks were predominant. The offer of fresh or minimally processed foods and beverages was reduced between the years 2019 and 2022. Although the supply of such food items was reduced in that period, almost all types of ultra-processed foods were present in more than half of the establishments. There was an increase in the average price of almost all evaluated foods; however, the only non-commercial establishment present on campus, the university restaurant, maintained the same low price for their meals.

In general, it was even more difficult to access healthy options in the food environment of 2022 owing to the frequent supply of ultra-processed foods that were less expensive than fresh or minimally processed foods. However, although there is only one university restaurant on campus, it plays a fundamental role in the university food environment. In addition to maintaining the same price for meals (which are subsidized by the university), the Restaurant serves about 3500 people daily, providing two meals, with a high-quality menu, aligned with the recommendations of the Dietary Guidelines for the Brazilian Population. For example, it provides legumes, fruits, and vegetables on a daily basis [23]. Such restaurants can be considered as food and nutritional security equipment within universities.

Characteristics of the food environment such as food availability, and physical and financial access to food, were affected by the COVID-19 pandemic. The assessment of the food environment showed worse availability of healthy foods, since some establishments closed down while the existing ones continued to offer a great supply of ultra-processed foods. A previous evaluation carried out in the same university campus, between 2011 and 2016, found a trend for an increase in the number of establishments in the period, especially those that offered unhealthy foods [10]. Together, these results point to an increase in the number of establishments in the university food environment that do not promote healthy eating. Our results are also in line with those of other studies conducted before the pandemic in different countries [7,8,9,10,11,13,14,24], which point to the predominance of establishments that mostly sold ultra-processed foods.

A study conducted at an Australian university also reported the predominance of unhealthy items in the university food environment [25]. At the University of Ghana, there was a predominance of establishments that sold unhealthy foods compared to those that sold healthy foods (50.7% vs. 39.9%) [26]. In Norway, the analysis of the food environment of an Oslo university showed the wide availability of ultra-processed foods [27]. Although all studies were conducted after the pandemic, the authors did not discuss the possible influences of the pandemic on university food environments, which indicates the innovative character of the analyses made in the present study.

The pandemic has had unprecedented effects on food systems around the world, impacting the way food is distributed, sold, obtained, prepared, and consumed [18]. The impacts of this health crisis caused changes in the food environment, especially in the availability of establishments (closedown of restaurants, suspension of classes at universities and schools), and limited financial access to healthy foods (increase in prices and reduction of purchasing power) [13,19,20]. Also, such impacts favored the sale of ultra-processed foods, since many companies promoted them based on their longer shelf life and ease of storage [20].

A survey conducted in Brazil described a worrying scenario regarding the prices of essential foods, such as rice, beans, and coffee powder, since the beginning of the pandemic. Prices increased in all 17 study capitals, and the highest annual increase occurred in 2020. This phenomenon further restricted the population’s access to basic and healthy foods [13,28].

Another extremely relevant data is related to food price inflation during the pandemic, occurring between 2020 and 2021. During this period, there was a remarkable increase of 23.2% in the Food and Beverage Price Index (IPAB), in addition to a 15.0% increase in the Broad Consumer Price Index (IPCA). This price increase has a direct impact on people’s ability to purchase the necessary amount of food; as a result, they may purchase an insufficient amount or replace healthy foods with other options that have lower nutritional quality [29]. Together, the reduced purchasing power of households and the increased food costs have led to significant consequences for food consumption in recent years [30]. This scenario poses serious challenges and requires a comprehensive approach to ensure that the population has access to quality food at affordable prices. This is also important in the context of universities, where students and workers remain for many hours a day.

The pandemic caused the country’s universities to reduce face-to-face activities and carry out only the essential ones. Thus, the commercial establishments and the university restaurant on the main campus suspended their services. In this context, tenants/managers of the commercial establishments located on campus, including snack bars, canteens, and restaurants, faced a sudden interruption in their economic activities and suffered a profound economic impact caused by the COVID-19 pandemic.

After the pandemic, about 30% of establishments that were operating at the end of 2019 were no longer operating in 2022. This situation was due, at least in part, to the financial instability faced by the managers of these establishments, which resulted in the termination of contracts with the university. These results converge with those of studies that analyzed the impact of the pandemic on the food retail environment in several locations, including three municipalities in the metropolitan region of Belo Horizonte [31], and in the city of New York [32,33]. In New York, approximately 35% of establishments operating in 2019 were no longer operating in 2020. This significant decrease can be largely attributed to physical distancing measures adopted during the pandemic, which probably contributed to the financial instability faced by the establishments [32,33].

In the scenario of the school food environment during the pandemic, school canteens also closed down, a situation similar to the one occurring at universities. In addition, when face-to-face activities were resumed, there may have been significant changes both in the management of canteens and at the points of sale of food, as well as in the type of food available. These changes could have been a window of opportunity to consider strategies not only to reverse but also to improve the food environment in schools when face-to-face activities were resumed [34]. However, in the case of the study university, this opportunity to improve food quality and types of establishments was not taken to promote healthier food environments.

In Brazil, there are no national or subnational standards, so each university can establish its rules. In the case of the university assessed, the rules are restricted to the non-commercial establishment, the university restaurant, does not affect contract agreements with commercial establishments. Elements raised in this study, combined with the recommendations of the Brazilian Dietary Guidelines can contribute to a proposal of healthy food standards and policies for the university food environment, as well as for their future evaluation, inspired by what was made for public universities from North Carolina, USA [35].

As mentioned above, this study makes an innovative analysis of comparison of important dimensions of the university food environment (availability and affordability) measured with an instrument validated for this context. Also, it is worth highlighting its pioneering way of comparing the university food environment before and after the pandemic, providing valuable information that can be used for monitoring and developing interventions for the sake of improving this scenario. Monitoring the university food environment can provide further insights into changes in such environment, can serve as a basis for future research and actions to promote healthy food initiatives, and reinforces the importance of university restaurants. Future studies can advance the proposal of synthesis indicators that contribute to the evaluation, monitoring, and advice for initiatives in food environments, as exemplified in a study carried out in Australia [36].

Despite the strengths pointed out, this study has some limitations that should be mentioned. First, this study focused exclusively on the university’s main campus, which does not reflect the diversity of contexts present on other campuses. This geographical restriction can limit the generalization of results for the entire university and other universities. In addition, data were collected for one month only; therefore, seasonal variations throughout the year could not be analyzed. These seasonal fluctuations can significantly impact food availability and variety, as well as prices, and this temporal variability needs to be considered when interpreting the results. However, data collection was conducted in the same period of the semester in both years, which can minimize differences due to the seasonality. Another limitation of this study is its exclusive focus on the university food environment, without accounting for the influence of the external environment (surroundings). This methodological decision was made to highlight aspects of the university food environment that could be under the control or supervision of the institution. Off-campus food options, along with economic and social factors that affect access to these options, can shape students’ dietary choices and habits. By concentrating solely on on-campus conditions, the study may not fully capture the broader impact of external factors, which also is dynamic. Future research could enhance this analysis by integrating both on- and off-campus food environments and also assessing the association between the university food environment and food consumption of the university community, offering a more comprehensive view of the challenges faced in maintaining a healthy diet and contributing to the nutrition and food security in the university.

## 5. Conclusions

The COVID-19 pandemic seems to have deteriorated the availability and affordability of food in the university food environment. Still, in general, before and after the pandemic, the environment discouraged healthy eating and/or stimulated unhealthy food choices by students and the university community. This is evidenced by the predominant availability of ultra-processed foods to the detriment of the supply of fresh or minimally processed foods and beverages. The COVID-19 pandemic has aggravated this situation, resulting in the closure of establishments, and increasing the difficulty of access to healthy options. Prices of most foods increased after the pandemic, except at the university restaurant, which stands out for offering healthy meals and promoting food and nutritional security in the university community. These findings point to the need for improvements in the university food environment, which can made through regulation of the sale and advertising of ultra-processed foods and by thoroughly reinforcing the role of the university restaurant in improving the food environment.

## Figures and Tables

**Table 1 ijerph-21-01544-t001:** General characteristics of commercial and non-commercial establishments on the main campus of the university.

General Characteristics of Establishments	Year					
	2019 (n = 20)			2022 (n = 14)		
	n	%	95% CI	n	%	95% CI
*Type of food served in the establishments **						
Snack	15	75.0	54.2–95.8	11	78.6	47.3–93.7
Meal	9	45.0	21.1–68.9	9	64.3	35.0–85.7
Convenience items	18	90.0	75.6–104.4	12	85.7	53.5–96.9
*Payment methods*						
Cash	19	95.0	84.5–105.5	13	92.9	58.0–99.2
Meal vouchers	4	20.0	0.8–39.2	1	7.1	0.8–42.0
Debit card	18	90.0	75.6–104.4	13	92.9	58.0–99.2
Credit card	10	50.0	26.0–74.0	14	100.0	-
Other	7	35.0	12.1–57.9	13	92.9	58.0–99.2

* The same establishment can offer more than one type of food.

**Table 2 ijerph-21-01544-t002:** Availability of food, beverages, and food preparations in commercial and non-commercial establishments of the main campus of the university.

Availability of Food, Beverages, and Food Preparations	Year					
	2019 (n = 20)			2022 (n = 14)		
	n	%	95% CI	n	%	95% CI
*Fresh or minimally processed foods*						
Raw vegetables	7	35.0	12.1–57.9	4	28.6	10.0–58.9
Cooked vegetables	7	35.0	12.1–57.9	2	14.3	3.1–46.5
Whole grain rice	6	30.0	8.0–52.0	2	14.3	3.1–46.5
Beans	8	40.0	16.5–63.5	4	28.6	10.0–58.9
Fruit or fruit salad	6	30.0	8.0–52.0	2	14.3	3.1–46.5
*Fresh or minimally processed beverages*						
Water	18	90.0	75.6–104.4	11	78.6	47.3–93.7
Fresh or prepared natural juices from pulp	11	55.0	31.1–78.9	5	35.7	14.3–65.0
*Ultra-processed foods*						
Sandwich or crepe	15	75.0	54.2–95.8	9	64.3	35.0–85.7
Fried/baked snack	16	80.0	60.8–99.2	11	78.6	47.3–93.7
Bonbons and chocolate bars	16	80.0	60.8–99.2	10	71.4	41.1–90.0
Candy (including gum)	18	90.0	75.6–104.4	11	78.6	47.3–93.7
Cereal bars	14	70.0	48.0–92.0	8	57.1	29.3–81.1
Sweet biscuit with filling	17	85.0	67.9–102.1	8	57.1	29.3–81.1
Sweet biscuit without filling	15	75.0	54.2–95.8	8	57.1	29.3–81.1
Biscuits and/or “packet” salted snacks	17	85.0	67.9–102.1	7	50.0	24.0–76.0
Whole cookies	5	25.0	4.2–45.8	3	21.4	6.3–52.7
Other sweets (honey bread, cake, brigadeiro)	17	85.0	67.9–102.1	10	71.4	41.1–90.0
*Ultra-processed beverages*						
Refreshment/Industrialized fruit drinks	8	40.0	16.5–63.5	4	28.6	10.0–58.9
Coconut water	5	25.0	4.2–45.8	3	21.4	6.3–52.7
Soft drinks (including flavored water)	17	85.0	67.9–102.1	11	78.6	47.3–93.7
Industrialized beverages made of fruit juices or fruit nectar	14	70.0	48.0–92.0	5	35.7	14.3–65.0
Ready-to-drink tea (e.g., ice tea, mate)	18	90.0	75.6–104.4	11	78.6	47.3–93.7
Isotonic and sports drinks (e.g., Gatorade)	5	25.0	4.2–45.8	6	42.9	18.9–70.7
Energy drinks (e.g., Redbull)	8	40.0	16.5–63.5	2	14.3	3.1–46.5
Guarana drink	18	90.0	75.6–104.4	9	64.3	35.0–85.7
Flavored milk or milk-based beverages or industrialized yogurt	15	75.0	54.2–95.8	6	42.9	18.9–70.7
Coffee (self-service machine)	16	80.0	60.8–99.2	4	28.6	10.0–58.9

**Table 3 ijerph-21-01544-t003:** Promotion and price in commercial and non-commercial establishments on the main campus of the university.

	Year					
Promotion	2019 (n = 20)			2022 (n = 14)		
	n	%	95% CI	n	%	95% CI
*Healthy substitution*						
French fries for vegetables	1	5.0	(−5.5–15.5)	0	0.0	-
White rice for whole grain rice	1	5.0	(−5.5–15.5)	0	0.0	-
White bread for whole bread	7	35.0	(12.1–57.9)	0	0.0	-
Soft drink for fresh or prepared natural juices from pulp	5	25.0	(4.2–45.8)	0	0.0	-
*Change portion size*						
Larger portions of food and drinks	8	40.0	(16.5–63.5)	0	0.0	-
Same price	5	25.0	(4.2–45.8)	0	0.0	-
Lower price	2	10.0	(−4.4–24.4)	0	0.0	-
Higher price	1	5.0	(−5.5–15.5)	0	0.0	-
Reduced portion of some item	8	40.0	(16.5–63.5)	0	0.0	-
Same price	6	30.0	(8.0–52.0)	0	0.0	-
Lower price	2	10.0	(−4.4–24.4)	0	0.0	-
*Combo/Promotion Presence*	14	70.0	(48.0–92.0)	9	64.3	(35.0–85.7)
**Price (R$)**	**Mean**	**Min-Max**		**Mean**	**Min-Max**	
Standard dish	8.33	2.00–13.00		12.67	2.00–15.99	
Meal by weight	39.75	34.90–43.99		-	-	
French fries	7.50	5.00–10.00		-	-	
Fruit or fruit salad	5.66	2.00–10.00		15.00	15.00–15.00	
Sweet or dessert	2.91	1.50–5.00		5.45	3.00–12.50	
Confectionery	0.60	0.10–3.00		1.16	0.20–2.50	
Cereal bars	1.72	1.00–3.00		1.93	1.50–2.00	
Biscuits and/or “packet” salted snacks	2.80	1.00–4.00		2.92	1.50–6.50	
Whole cookies	3.30	1.00–4.50		2.10	1.00–4.00	
Sweet biscuit without filling	3.10	1.00–4.00		4.76	2.00–6.50	
Sweet biscuit with filling	3.41	1.00–5.00		3.58	1.30–6.50	
Sandwich	5.22	3.50–6.90		5.67	3.00–8.00	
Fried snacks	3.86	3.50–4.50		4.88	3.00–6.00	
Baked snack	3.90	2.00–5.00		4.82	2.50–6.00	
Soft drinks	3.04	1.80–5.50		4.36	3.00–6.00	
Refreshment/industrialized fruit drinks/guarana drink/ready-to-drink tea	1.83	1.00–4.00		2.10	1.00–4.00	
Fresh or prepared natural juices from pulp	5.04	3.50–7.00		6.50	6.00–7.00	
Fruit nectar	4.03	2.00–6.00		5.08	3.00–6.80	
Industrialized beverages made of fruit juices	2.83	2.00–4.50		3.0	3.00–3.00	
Water	2.33	2.00–3.20		2.30	2.00–3.00	
Isotonic and sports drinks	5.90	5.50–6.00		7.80	6.00–10.00	
Energy drinks	7.25	6.00–10.00		10.50	10.00–11.00	
Flavored milk or milk-based beverages or industrialized yogurt	3.24	2.50–4.00		3.93	3.50–4.50	
Mixed milk and fruit beverage	5.68	4.50–7.00		-	-	

## Data Availability

The datasets used and/or analyzed during the current study are available from the corresponding author upon reasonable request.

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
