# Peer review of "Changes in Availability and Affordability on the University Food Environment: The Potential Influence of the COVID-19 Pandemic"

_ijerph, 2024, doi:10.3390/ijerph21121544_

Round 1

Reviewer 1 Report

Comments and Suggestions for Authors

The article is clear and well-organized. It highlights the importance of ongoing surveillance in organizational food environments. Additionally, it contributes by detailing the direct impacts of the COVID-19 pandemic on how these food environments are structured.

I have a few minor suggestions:

 -Abstract

In the abstract, the last sentence seems to indicate that exacerbation of the university food environment is inherently harmful. I suggest making this explicit: "The pandemic seems to have negatively exacerbated the university food environment, making healthy choices harder." (Lines 21-22).

 -Table 1

Could you explain what the "other" payment methods are? Their growth is significant and parallels the increasing utilization of credit cards.

-Discussion

Consider the impact of the food environment outside the university as a topic for discussion or as a limitation in the study.

Author Response

The article is clear and well-organized. It highlights the importance of ongoing surveillance in organizational food environments. Additionally, it contributes by detailing the direct impacts of the COVID-19 pandemic on how these food environments are structured.

I have a few minor suggestions:

-Abstract

In the abstract, the last sentence seems to indicate that exacerbation of the university food environment is inherently harmful. I suggest making this explicit: "The pandemic seems to have negatively exacerbated the university food environment, making healthy choices harder." (Lines 21-22).

Response: Thank you for your attention to enhancing the text. We have changed the word “exacerbated” to “deteriorated” to clarify the meaning and make the conclusion more explicit (lines 22-23).

The pandemic seems to have deteriorated two important dimensions of food access in the university food environment, making healthy choices harder for students and the university community.

-Table 1

Could you explain what the "other" payment methods are? Their growth is significant and parallels the increasing utilization of credit cards.

Response: Thank you for pointing it out. We have added the following sentence (lines 136-138):

Other payment methods, such as Pix, gained prominence in 2022. Pix is an instant payment system developed by the Central Bank of Brazil in 2020.  (Table 1).

We also added a footnote:

The term Pix is not a traditional acronym with each letter representing a specific word; instead, it was selected by the Central Bank of Brazil to convey the concept of instant payments.

-Discussion

Consider the impact of the food environment outside the university as a topic for discussion or as a limitation in the study.

Response: Thank you for your suggestion. It was added to study limitations (lines 299-310).

Another limitation of this study is its exclusive focus on the university food environment, without accounting for the influence of the external environment (surroundings). This methodological decision was made to highlight aspects of the university food environment that could be under the control or supervision of the institution. Off-campus food options, along with economic and social factors that affect access to these options, can shape students' dietary choices and habits. By concentrating solely on on-campus conditions, the study may not fully capture the broader impact of external factors, which also is dynamic. Future research could enhance this analysis by integrating both on- and off-campus food environments and also assessing the association between the university food environment and food consumption of the university community, offering a more comprehensive view of the challenges faced in maintaining a healthy diet and contributing to the nutrition and food security in the university.

Reviewer 2 Report

Comments and Suggestions for Authors

Dear Authors,

The study aimed to describe potential changes that occurred between 2019 and 2022 in the availability and affordability of food and beverages in the food environment of a Brazilian public university.

However, there are several issues.
1. the study tried to describe the potential changes because of the covid-19, but I have not yet found the urgency of this study purpose research question. if you need to explain to the reader about the effect of the food habit because of the changes of food availability or affordability then your data analysis need more improvement than the univariate analysis.

2. this study would be more benefit to the policy makers if the food availability and affordability could really shown that it affect the nutrition security or the nutrition status of the university students.

Author Response

The study aimed to describe potential changes that occurred between 2019 and 2022 in the availability and affordability of food and beverages in the food environment of a Brazilian public university. However, there are several issues.

  1. the study tried to describe the potential changes because of the covid-19, but I have not yet found the urgency of this study purpose research question. if you need to explain to the reader about the effect of the food habit because of the changes of food availability or affordability then your data analysis need more improvement than the univariate analysis.

Response: Thank you for your comment and the opportunity to clarify our analytical approach. We chose univariate analysis as the primary goal of this study is to provide an initial, exploratory description of changes in food availability and accessibility in the university setting before and after the pandemic. Since this research does not aim to establish causal relationships between food availability and dietary habits but rather to offer a general overview of the food environment, univariate analysis is sufficient to fulfill the study’s objectives. This approach enables us to capture and illustrate variations in the studied items clearly and directly, without requiring more complex statistical inferences that could shift the focus from objectively describing the observed changes.

Additionally, a more detailed analysis would likely necessitate extra data and variables that were not included in this initial descriptive study, as they were outside the scope of this research. For these reasons, we believe that univariate analysis adequately supports the intended goal of describing the transformations in the university food environment during the pandemic period.

We made an inclusion in the Discussion, highlighting the importance of future studies focusing on associations (lines 306-310).

Future research could enhance this analysis by integrating both on- and off-campus food environments and also assessing the association between the university food environment and food consumption of the university community, offering a more comprehensive view of the challenges faced in maintaining a healthy diet and contributing to the nutrition and food security in the university.

  1. this study would be more benefit to the policy makers if the food availability and affordability could really shown that it affect the nutrition security or the nutrition status of the university students.

Response: We appreciate the feedback and recognize the importance of demonstrating how food availability and accessibility directly impact the nutritional security and health status of university students, especially to inform more effective policy decisions. This study, however, was designed as an initial descriptive analysis of the university food environment, aiming to map changes in food availability and accessibility during the pandemic period. Although it does not directly assess the nutritional effects on students, we believe that providing an overview of the food environment is a crucial foundational step for deeper investigations into health and nutrition impacts.

We acknowledge that future studies incorporating data on students' nutritional status and food security could further strengthen the evidence needed to support public policies promoting healthier and more accessible food environments. In this way, this study serves as an exploratory basis for a broader and more well-rounded understanding, laying the groundwork for subsequent research that more directly addresses policymakers' needs.

We also made an inclusion in the Discussion about future studies (lines 306-310).

Future research could enhance this analysis by integrating both on- and off-campus food environments and also assessing the association between the university food environment and food consumption of the university community, offering a more comprehensive view of the challenges faced in maintaining a healthy diet and contributing to the nutrition and food security in the university.

Reviewer 3 Report

Comments and Suggestions for Authors

The article basis is an analysis of how food environment has changed since 2019 pandemic in a Brazilian university.

My comments about the article are the following:

-       Line 2, title: The word food is repeated twice in the title. Review the possibility of changing it, for example to Changes in availability and affordability on the university food environment: the potential influence of the COVID-19 pandemic

-       Line 10: change “food” for “eating habits”, “food intake” or another related concept.

-       Line 21: Explain the meaning of the word “exacerbated”. That it means that the food environment has become worse?

-       Line 38: Line 10: change “food” for “eating habits” or another related concept.

-       Line 41: review the references citations. Change [6-14], both here and in the rest of the article.

-       Line 64: Try not repeating the word “main” twice in the same sentence

-       Table 1: include the statistical significance data in the table of the different parameters in the table, apart from mentioning them in the text.

-       Table 1: Why is the credit card 2022 95% CI not included?

-       Lines 142-161: It would be more adequate to compare the total number of establishments that offer the different categories of food, not the percentage.

-       Line 201: In 2002?

-       Line 232: This idea is repetitive with the previous paragraph. Try to combine both paragraphs.

-       Line 264: Explain which authors are you speaking about.

Conclusions: Evaluate the possibility of mentioning the relevance of reinforcing the paper of the university restaurant for improving the food environment.

References: Review references 13 and 21, are they the same?

Author Response

The article basis is an analysis of how food environment has changed since 2019 pandemic in a Brazilian university.

My comments about the article are the following:

-Line 2, title: The word food is repeated twice in the title. Review the possibility of changing it, for example to Changes in availability and affordability on the university food environment: the potential influence of the COVID-19 pandemic

Response: We agree with the suggestion. The change was made and now the title is: Changes in availability and affordability on the university food environment: the potential influence of the COVID-19 pandemic

-Line 10: change “food” for “eating habits”, “food intake” or another related concept.

Response: We agree with the suggestion (line 10). “Food” was changed to “eating habits”.

-Line 21: Explain the meaning of the word “exacerbated”. That it means that the food environment has become worse?

Response: We have changed the word “exacerbated” to “deteriorated” to make the meaning clearer (lines 21-23).

The pandemic seems to have deteriorated two important dimensions of food access in the university food environment, making healthy choices harder for students and the university community.

-Line 38: Line 10: change “food” for “eating habits” or another related concept.

Response: Thank you for the observation. We would like to clarify that, in this context, the term 'food' refers to food provision rather than students' dietary habits. The intention is to emphasize that, upon entering university, many students take on the responsibility of managing their own food, along with housing, budgeting, and time management.

-Line 41: review the references citations. Change [6-14], both here and in the rest of the article.

Response: We agree with the request to simplify sequential reference citations and we implemented the suggested format here and in the rest of the article.

-Line 64: Try not repeating the word “main” twice in the same sentence

Response: We have changed “main building to “major building” (line 65).

The university has 11 campuses, but this study focuses on the main campus, whose major building has an urban and vertical design, and 12 floors.

-Table 1: include the statistical significance data in the table of the different parameters in the table, apart from mentioning them in the text.

Response: Thank you for your comment. We have chosen to use the 95%CI approach instead of hypothesis tests and their p-values. Nowadays, there is an important body of medical and epidemiological literature pointing out the overemphasis on the use of p-values to dichotomize significant or non-significant results, which can lead to misleading interpretations of the results. In this regard, the use of CI allows determining the size of the difference of measures between groups, rather than simply indicating whether there is (or not) a statistically significant difference. In the case of Table 1, no statistically significant differences between the parameters analyzed were identified. For this reason, we have not included additional information of statistical significance in the table, although the full results are discussed in the text.

- Table 1: Why is the credit card 2022 95% CI not included?

Response: Thank you for the observation. The 95% confidence interval (CI) for credit card use in 2022 was not included in the table due to insufficient variability in the data. As all surveyed establishments accepted credit cards, a confidence interval calculation was not applicable. The uniform acceptance rate across establishments makes a confidence interval unnecessary for this category.

-Lines 142-161: It would be more adequate to compare the total number of establishments that offer the different categories of food, not the percentage.

Response: We presented both absolute and relative frequencies in Tables but chose to compare the data as percentages rather than absolute numbers to make it easier to compare food categories, as percentages offer a clearer view of relative proportions regardless of changes in the total number of establishments over time. This approach allows for more intuitive comparisons, minimizing the impact of absolute differences that might complicate the analysis.

-Line 201: In 2002?

Response: Sorry, but we couldn’t identify this typo in our manuscript.

-Line 232: This idea is repetitive with the previous paragraph. Try to combine both paragraphs.

Response: Thank you for the observation. We would like to clarify that the paragraphs are not repetitive; rather, they address complementary aspects of the food landscape during the pandemic in Brazil. The first paragraph highlights the rise in prices of essential foods across various Brazilian capitals, focusing specifically on the impact in 2020 and illustrating how this increase has hindered access to basic, healthy foods. The second paragraph expands the analysis to cover food inflation from 2020 to 2021, providing data on the rise in the Food and Beverage Price Index (IPAB) and the Broad Consumer Price Index (IPCA), and discussing the direct impact of these increases on families’ purchasing power, including within universities.

-Line 264: Explain which authors are you speaking about.

Response: Thank you for pointing it out. The sentence was changed to improve clarity (lines 265-267).

These changes could have been a window of opportunity to consider strategies not only to reverse, but also to improve the food environment in schools when face-to-face activities were resumed.

Conclusions: Evaluate the possibility of mentioning the relevance of reinforcing the paper of the university restaurant for improving the food environment.

Response: Thank you for the suggestion. It was included in the conclusion section (lines 322-325).

These findings point to the need for improvements in the university food environment, which can made through regulation of the sale and advertising of ultra-processed foods and by thoroughly reinforcing the role of the university restaurant in improving the food environment.

References: Review references 13 and 21, are they the same?

Response: Thank you for the observation. We confirm that references 13 and 21 are indeed duplicates due to an entry error. The correct reference for citation 21 is (lines 391-392):

Botelho LV, Cardoso LO, Canella DS. COVID-19 and the digital food environment in Brazil: reflections on the pandemic’s influence on the use of food delivery apps. Cad Saude Publica. 2020;36:e00148020, 2020.

We proceeded with the replacement as suggested.

Once again, we appreciate your careful attention and guidance, which have significantly enhanced this study.

Round 2

Reviewer 2 Report

Comments and Suggestions for Authors

Dear authors,

in the abstract, it's written ".., dropping from 20 to 14,..."
plase add the unit of measurement of these numbers.

The conclusion is still vague and have not yet represented the content of the manuscript.

-processed food

Author Response

In the abstract, it's written ".., dropping from 20 to 14,..." please add the unit of measurement of these numbers.

Response: We rewrote the sentence to make it clear that it refers to absolute numbers.

Over the period, there was a decrease in the on-campus number of establishments, dropping from 20 to 14.

The conclusion is still vague and have not yet represented the content of the manuscript.

Response: We reviewed both conclusions, of the abstract and the article.

Abstract:

The pandemic seems to have deteriorated the availability of healthy food and their prices in the university food environment, making healthy choices harder for students and the university community.

Article:

The COVID-19 pandemic seems to have deteriorated the availability and affordability of food in the university food environment. Still, in general, before and after the pandemic, the environment discouraged healthy eating and/or stimulated unhealthy food choices by students and the university community.